# Melatonin from Microorganisms, Algae, and Plants as Possible Alternatives to Synthetic Melatonin

**DOI:** 10.3390/metabo13010072

**Published:** 2023-01-02

**Authors:** Marino B. Arnao, Manuela Giraldo-Acosta, Ana Castejón-Castillejo, Marta Losada-Lorán, Pablo Sánchez-Herrerías, Amina El Mihyaoui, Antonio Cano, Josefa Hernández-Ruiz

**Affiliations:** Phytohormones & Plant Development Laboratory, Department of Plant Biology (Plant Physiology), Faculty of Biology, University of Murcia, 30100 Murcia, Spain

**Keywords:** dietary supplements, GMOs, melatonin, microorganisms, phytomelatonin, plant raw material

## Abstract

Melatonin dietary supplements are widely consumed worldwide, with developed countries as the largest consumers, with an estimated annual growth rate of approximately 10% until 2027, mainly in developing countries. The wide use of melatonin against sleep disorders and particular problems, such as jet lag, has been added to other applications, such as anti-aging, anti-stress, immune system activation, anticancer, and others, which have triggered its use, normally without a prescription. The chemical industry currently covers 100% of the needs of the melatonin market. Motivated by sectors with more natural consumption habits, a few years ago, the possibility of obtaining melatonin from plants, called phytomelatonin, arose. More recently, the pharmaceutical industry has developed genetically modified microorganisms whose ability to produce biological melatonin in bioreactors has been enhanced. This paper reviews the aspects of the chemical and biological synthesis of melatonin for human consumption, mainly as dietary supplements. The pros and cons of obtaining melatonin from microorganisms and phytomelatonin from plants and algae are analyzed, as well as the advantages of natural melatonin, avoiding unwanted chemical by-products from the chemical synthesis of melatonin. Finally, the economic and quality aspects of these new products, some of which are already marketed, are analyzed.

## 1. Introduction

Melatonin (*N*-acetyl-5-methoxytryptamine) is widely used around the world as a dietary supplement. In general, melatonin is used as a sleep aid supplement, a mild tranquilizer, a generalist antioxidant, and an anticancer and anti-aging component, among others [1]. According to the American Psychiatric Association (APA), approximately one-third of adults suffer from insomnia during their lifetime [2]. It manifests itself in incessant problems falling asleep and staying asleep. Therefore, it is very likely that the use of synthetic melatonin will spread. In 2019, the global production of synthetic melatonin, which was around 4000 tons, accounted for around 1.3 billion USD. This vast market is fully assisted by chemical melatonin, whose synthesis process is very cheap, effective, and, therefore, lucrative. The melatonin market is expected to grow at a CAGR (compound annual growth rate) of >10% over the next 5 years. With this considerable increase in demand, the insomnia problems generated by the COVID-19 pandemic have been of great relevance [3]. North America has the highest consumption by far, followed by Europe. The global melatonin market is mainly controlled by a few major companies, such as BASF, Aspen Pharmacare Australia, Nature’s Bounty, Pfizer Inc., Natrol LLC, Aurobindo Pharma, and Biotics Research Co. Note that the consumption of melatonin for medical purposes involves about 50% of the synthetic melatonin produced; the rest has chemical and industrial applications [2,4].

Biologically, melatonin is a molecule widely distributed in all kingdoms of living organisms [5]. Discovered in 1958 in the pineal gland of a cow [6] and later in humans [7], it is one of the most studied biomolecules, and its multiple functions are known, mainly in mammals [8,9], but also in fish [10,11,12], poultry [13,14], and invertebrates [15]. In animal and human cells, melatonin acts as an antioxidant—a relevant role attributed to it in 1993 [16,17,18]. Melatonin acts as an interesting cell protector in stressful situations, in various physiological aspects in humans, and, according to multiple studies, benefits an improvement in different diseases and dysfunctions. Figure 1 shows some of the protective and regulatory actions of melatonin in humans and presents melatonin as an interesting pleiotropic molecule, standing out due to its relevance, the role of melatonin in the regulation of lipid and glucose metabolism, inducing nocturnal insulin resistance and diurnal insulin sensitivity. This effect seems to be associated with nocturnal fasting and diurnal feeding, preventing excessive weight gain [19]. We also highlight its role as an anti-oncogenic agent, inhibiting the growth, proliferation, and metastasis of several tumors. The treatment of tumors with melatonin improved chemo- and radiotherapy sensitivity, acting as a synergistic molecule in the control of cancer cells. Additionally, melatonin mitigates acute damage to normal cells, protecting them against drug toxicity, possibly by enhancing immune responses [20,21,22]. Among the dysfunctions and diseases where the beneficial effects of melatonin have been studied are neurological ones, such as Alzheimer’s, Parkinson’s, fibromyalgia, depression, attention-deficit hyperactivity disorder, autism, and migraines; cardiovascular health problems, including hypercholesterolemia, hypertension, metabolic syndrome, and glycemic imbalance; gastrointestinal health problems, such as gastroesophageal reflux, ulcers, and irritable bowel syndrome; immunological health problems, such as multiple sclerosis, autoimmune responses (athletic stress, toxic stress, psoriasis, etc.), sepsis, COVID-19, etc. [3,23,24,25,26]; and also osteopenia [27], sarcopenia [28], pre-eclampsia, fertility, polycystic ovarian syndrome, and menopause, among others [29,30,31,32]. However, even though melatonin is a molecule that has been widely studied since the 1950s, the studies carried out require more clinical and extensive double-blind trials in order to clarify its sometimes confusing pleiotropic action [33,34].

However, melatonin is well known for being the hormone that regulates sleep. Its oscillating levels in the blood flow according to the periods of light and darkness (circadian rhythms) due to the release of melatonin by the pineal gland is one of the most studied and known aspects of this molecule. The increase in blood melatonin levels during the first period of sleep to around 150–220 pmoles/mL acts on sleep initiation, reduces sleep latency and fragmentation, and increases sleep duration and quality [1,35,36]. Melatonin acts as an internal synchronizer of the circadian sleep–wake cycle and seasonal rhythmicity. In this sense, many sleep disorders have been treated with melatonin, including delayed sleep phase syndrome, night shift work sleep disorder, seasonal affective disorder, sleep disorders in the blind and aging, and pathophysiological disorders of children, with notable improvements in sleep quality [37,38,39,40,41]. The most widespread disorder treated with melatonin is jet lag—a de-phasing in sleep–wake rhythms following trans-oceanic flights [42,43,44,45]. Possibly, the emphasis in studies on its role as a sleep regulator has caused a lack of studies on its possible role in many other physiological and clinical aspects.

Melatonin in plants, so-called phytomelatonin, was discovered simultaneously by three research groups in diverse plant material in 1995 [46,47,48]. The term phytomelatonin, which refers to melatonin of vegetable origin (plants and algae), is used to differentiate it from animal and/or synthetic melatonin. This term is very widespread and is used continuously in studies of phytochemistry, plant physiology, botany, food chemistry, etc., on plant melatonin. In plants, phytomelatonin is also a pleiotropic molecule, presenting multiple roles in diverse physiological responses (Figure 1). The regulation by melatonin of aspects such as photosynthesis, including stomatal CO_2_ uptake and water economy, carbohydrate, lipid, nitrogen, and sulfur metabolism, and simple phenol, flavonoid, and terpenoid metabolism, has demonstrated crucial interest in the basic and technical processes of vegetative (germination, plant growth, rooting, branching, etc.) and reproductive development, including fertility, parthenocarpy, seed and fruit development, ripening, senescence, and the conservation of fruits and cut flowers [49,50,51,52,53]. Generally, melatonin regulates these processes through the action of the plant hormone network, up/down-regulating several biosynthesis, catabolic, and transcription factors that are plant hormone-related [54,55,56]. One of the aspects of greatest agronomic and biotechnological interest is the role of phytomelatonin as a promoter of tolerance against biotic and abiotic stresses [57,58,59,60,61,62,63,64,65,66,67,68] (Figure 1). Currently, phytomelatonin is presented as an interesting eco-friendly tool to control biological diseases and to facilitate the resistance/adaptation of plants to/against climate change.

## 2. Biosynthesis of Melatonin

Melatonin is an acetylated compound derived from serotonin. Both indolic amines are synthesized from the amino acid tryptophan in a biosynthetic pathway that has been extensively studied in both animals and plants [69,70]. In plants, tryptophan is converted into tryptamine by the enzyme tryptophan decarboxylase (TDC) (Figure 2). Tryptamine is then converted into 5-hydroxytryptamine (serotonin) by tryptamine 5-hydroxylase (T5H), an enzyme that has been extensively studied in rice, and which could act with many substrates, although this has not been studied in depth. Serotonin is *N*-acetylated by serotonin *N*-acetyltransferase (SNAT). *N*-acetylserotonin is then methylated by acetylserotonin methyl transferase (ASMT)—a hydroxyindole-*O*-methyltransferase—which generates melatonin. In plants, the methylation of *N*-acetylserotonin can also be performed by caffeic acid *O*-methyltransferase (COMT), a class of enzyme that can act on a variety of substrates, including caffeic acid and quercetin [71]. Serotonin may also be transformed into 5-methoxytryptamine by ASMT/COMT to generate melatonin after the action of SNAT. This route would occur in senescence and/or stress situations [70,72]. Furthermore, melatonin can be generated through the formation of *N*-acetyltryptamine by SNAT, which would be converted into *N*-acetylserotonin by T5H [73], although this route has not been demonstrated, possibly because T5H is the least studied enzyme of the pathway (Figure 2). Interestingly, up to four genes encoding histone deacetylases (DAC) have been identified in rice plants that can reverse the steps from serotonin to *N*-acetylserotonin and from 5-methoxytryptamine to melatonin. DAC, expressed in chloroplast, exhibited enzyme activity toward *N*-acetylserotonin, *N*-acetyltryptamine, and melatonin, with the highest deacetylase activity for *N*-acetyltyramine [74].

In animal cells, serotonin is formed from 5-hydroxytryptophan after the sequential action of tryptophan hydroxylase (TPH) and TDC. Although TPH has not been detected in plants, the presence of 5-hydroxytryptophan suggested that some enzymatic activity, such as that of TPH, acts to a lesser extent in plant cells. Moreover, melatonin can be generated through the formation of 5-methoxytryptamine, mainly under stress conditions as proposed by several authors, suggesting that the melatonin biosynthesis pathway may follow various alternative routes compared with animal cells, with a greater ability to adapt to metabolic changes in plants [72,75]. All the named enzymes have been detected and characterized in rice and Arabidopsis, except TPH, which is well known in animals but not in plants. Nevertheless, some authors have proposed that T5H can act as a hydroxylase with low substrate specificity and is capable of acting in all the hydroxylation steps described [70,76,77,78,79]. This same broad substrate specificity can also be attributed to SNAT, ASMT, and COMT enzymes. Melatonin intermediates are produced in various subcellular compartments, such as the cytoplasm, endoplasmic reticulum, mitochondria, and chloroplasts, which determine the subsequent enzymatic steps [80,81].

In microorganisms, there are few studies on the melatonin biosynthesis pathway [82]. *Saccharomyces* and bacteria (*Geobacillus*, *Bacillus*, and *Pseudomonas*) produced both serotonin and melatonin at different concentrations [83,84,85,86,87,88,89]. Moreover, the production of melatonin was evidenced by other authors in the cultures of the yeasts *Pichia kluyveri*, *Saccharomyces cerevisiae*, and *S. uvarum* and bacteria (*Agrobacterium*, *Pseudomonas*, *Variovorax*, *Bacillus*, and *Oenococcus*) [85,90,91] and previously in the photosynthetic bacteria *Rhodospirillum rubrum* [92] and *Erythrobacter longus* [93] and *Escherichia coli* [94].

In the yeast *Saccharomyces cerevisiae*, unlike in plants and animals, it seems that the biosynthesis of 5-hydroxytryptophan from tryptophan does not occur. Interestingly, several of the described stages appear to be reversible in *S. cerevisiae*, such as between 5-hydroxytryptophan and serotonin, *N*-acetylserotonin and melatonin, and 5-methoxytryptamine and melatonin [90,95], as detailed in Figure 2. In *Bacillus amyloliquefaciens* SB-9 and *Pseudomonas fluorescens* RG11, 5-hydroxytryptophan, serotonin, and *N*-acetylserotonin, but not tryptamine, were detected [85,86]. So, several genes from bacterial origin were used to build a melatonin-producing *Escherichia coli* strain. For example, the DDC gene, encoding an aromatic L-amino acid decarboxylase from *Candidatus Koribacter versatilis* Ellin 345 and *Draconibacterium orientale*, and the AANAT gene, encoding an aralkylamine *N*-acetyltransferase from *Streptomyces griseofuscus*, were assayed [96,97]. Undoubtedly, further studies are needed to elucidate the complete biosynthetic pathways of melatonin in different prokaryotic and eukaryotic microbes [82].

## 3. Biological Melatonin versus Synthetic Melatonin

Initially, melatonin was obtained for experimental and clinical studies from animal sources (mainly from the pineal gland and urine), with the consequent risk of viral transmission [98,99]. These techniques were withdrawn when melatonin could be obtained by chemical synthesis [100]. Currently, all melatonin used for industrial and medical purposes is obtained by using chemical synthesis methods. These methods, which presented serious problems in the 1980s, including deaths due to the presence of by-products of synthesis from tryptophan [101], are much safer and more efficient today. However, melatonin preparations have described the presence of a whole set of undesirable by-products due to their toxic nature. Figure 3 shows three of the most commonly used chemical synthesis routes for melatonin and the by-products that are generated in its synthesis [102]. The synthesis of melatonin from tryptophan derivatives (Figure 3, Scheme A) generates toxic by-products that have sometimes caused significant diseases, such as eosinophilia myalgia syndrome [101,103,104], while the most current methods (Figure 3, Scheme B) for the synthesis of melatonin from phthalimide [105] raise important doubts about the toxicity of several of the by-products that are generated [106]. In addition, Fischer indole reactions from allylamine (Figure 3, Scheme C) present dangerous and toxic reactants [107].

On the other hand, obtaining melatonin from non-animal biological sources is presented as a strong commitment to the future, not to replace synthetic melatonin but to be a more natural complementary and alternative source [108].

## 4. Strategies to Obtain Biological Melatonin

Melatonin is present in all known biological species, from prokaryote to eukaryote, including yeasts, algae, fungi, and plants, as well as animals [80,109,110,111]. Below, the methodologies developed in microorganisms and plants as possible sources of natural melatonin are presented.

### 4.1. Melatonin from Microorganisms

a. Saccharomyces

The first approach to the production of biological melatonin has recently been made by a group of the Danish pharmaceutical company Novo Nordisk using genetically modified *Saccharomyces cerevisiae* (Table 1, product #1). Germann and co-workers constructed a recombinant melatonin pathway in a strain of yeast that contained heterologous genes encoding several melatonin biosynthesis enzyme and co-factor supporting pathways [112]. The transgenic yeast codified different genes from *Rattus norvegicus*, *Lactobacillus ruminis*, *Pseudomonas aeruginosa*, *Homo sapiens*, *Schistosoma mansoni*, *Bos Taurus*, and *Salmonella enterica*. Feeding yeasts only glucose and acetyl Co-A, melatonin production reached 14.5 mg·L^−1^ at 76 h. Nevertheless, according to other authors, some problems, such as high *N*-acetylserotonin accumulation in yeast cells, unbalanced gene expression, and the identification of some potential toxic intermediates, must be addressed [113].

b. *Escherichia coli*

In a second approach, this time using a transgenic-modified *Escherichia coli* culture (Table 1, product #2), Novo Nordisk reported the biological melatonin production from a heterologous strain constructed from recombinant *E. coli*, including several genes such as the TDC gene from *Candidatus Koribacter versatilis*, the SNAT gene from *Streptomyces griseofuscus*, and the human ASMT gene. Additionally, some tryptophan-related genes were blocked or deleted to prevent undesirable repression, degradation, and export transport [96,97]. After several strain improvements and feeding with mineral salts, vitamins, and antibiotics, the cultured cells generated melatonin at ~1 g·L^−1^ using glucose as the sole carbon source and up to 2 g·L^−1^ in tryptophan-fed cells, with negligible levels of by-products. Thus, according to the authors, these GMO *E. coli* strains may be the basis for future biologically commercial melatonin production using microbial cell factories. Nevertheless, the use of transgenic organisms to produce substances for human consumption can be problematic when the goal is to bring a natural product to a sensitized consumer or anti-GMOs consumer.

c. Lactic acid bacteria

Melatonin was also produced industrially by microbial fermentation, as reported in [114]. Melatonin biosynthesis was directed by multi-strain lactic acid bacteria, such as *Lactobacillus* sp. (*L. brevis*, *acidophilus*, *bulgaricus*, *casei subspec. sakei*, *fermentum*, *helveticus subspec. jogorti*, *plantarum*); *Bifidobacterium* sp. (*B. breve* spp. *breve*, *longum* spp. *infantis*); *Enterococcus* sp. (*E. faecalis TH10*); and *Streptococcus thermophilus*. The products manufactured under this technology are marketed by Quantum Nutrition Labs (Texas, USA) as Melatonin Drops, Qultured™, containing 8 mg of yeast melatonin (Table 1, product #3).

d. Chlorella

A product made from algae is Herbatonin^®^ (Table 1, product #4), formulated in pills containing 0.3 or 3 mg of phytomelatonin, although in Europe it is marketed in doses of 0.3 and 1.9 mg, according to EU laws. This formulation contains several plant species such as rice (*Oryza sativa* L.) and alfalfa (*Medicago sativa* L.), together with the green alga *Chlorella pyrenoidosa* and *C. vulgaris*. Our data show that these microalgae contained no more than 2–15 ng·g DW^−1^ [115], and the companion plant species contain very low levels of phytomelatonin, 1–5 ng·g^−1^ in rice and 16 ng·g^−1^ in alfalfa [116]. The presence of *Chlorella* suggests that the phytomelatonin is mainly obtained by culturing the green algae in bioreactors, possibly fed with precursors such as tryptophan, in a similar way to that in *Achillea millefolium* [117], although there are no published data on the method of obtaining these phytomelatonin-rich extracts, only their biochemical characterization [118]. There are also no data on the control of the presence of cyanotoxins in these extracts due to the possible contaminations by cyanobacteria (blue-green algae). These cyanotoxins have several unwanted effects related to carcinogenicity, hepatotoxicity, and neurotoxicity, among others. Thus, the detection of cyanotoxins in some algal dietary supplements reinforces the need for better quality control [119,120].

**Table 1 metabolites-13-00072-t001:** Most relevant biological melatonin products from microorganisms and plants and their trademarks or producers.

Product	Biological Origin	Natural Level	Concentrated at	Trademark	Ref.
1	*S. cerevisiae* (GMO)	85 µg·L^−1^	14.5 mg·L^−1^	- Novo Nordisk Co.	[112]
2	*E. coli* (GMO)	~ng·L^−1^	1–2 g·L^−1^	- Novo Nordisk Co.	[97]
3	*Lactobacillus* and others	ng-µg·L^−1^	8 mg in liquid (0.015%)	Melatonin Qultured * Quantum Nutrition Labs	[114]
4	*Chlorella* Rice Alfalfa	2–15 ng·g DW^−1^ 1–5 ng·g DW^−1^ 16 ng·g DW^−1^	0.3, 3 mg·pill (1.4%)	Herbatonin * Natural Health Int. Co.	[118]
5	Tart cherries	14 ng·g FW^−1^	14 µg·pill (0.003%)	Sleep Support * Tru2U Co.	[121]
6	Rice seed	7–216 ng·g DW^−1^	-	-	[122,123,124]
7	Mustard seed	194–660 ng·g DW^−1^	-	-	[125,126]
8	St. John’s Wort	See Table 2	0.3, 1.9 mg·pill (1%)	Mélatonine Végétale * Dynveo Co.	-
9	Valerian root	1.5–2 µg·g DW^−1^	-	- UMU	[127]
10	MAPs	1–20 µg·g DW^−1^	-	Bioriex UMU	[128,129]

* Commercialized product.

### 4.2. Melatonin from Plants

Obtaining melatonin from plants was, apparently, one of the most successful strategies. Phytomelatonin has been detected in all photosynthetic species (plants, algae, and some bacteria) analyzed so far. In algae and plants, endogenous levels of phytomelatonin are very low, between picograms and nanograms per gram of tissue [110,130]. On the other hand, phytomelatonin-rich extracts can have several advantages in addition to being natural, such as the presence of biologically healthy compounds, such as antioxidants, vitamins, etc. However, obtaining phytomelatonin-rich extracts in sufficient concentrations to meet the expectations of the natural supplement industry has not been an easy task due to contrary factors, such as low and variable content in phytomelatonin. Some of the plant products currently on the market or that have future possibilities are shown in Table 1.

a. Phytomelatonin from cherries

Phytomelatonin obtained from freeze-dried Montmorency tart cherry skin extracts was possibly the first phytomelatonin commercialized product (Sleep Support^®^, New Zealand) (Table 1, product #5). This product contains around 14 μg of phytomelatonin per pill, a very low amount to improve sleep quality but quite an achievement considering the very low amounts of phytomelatonin contained in the original plant material (~14 ng·g^−1^ fruit [121]. However, some studies suggest that cherry-rich products may have some effect on improving antioxidative status and sleep health [131,132].

b. Phytomelatonin from rice seeds

Based on studies of phytomelatonin extraction methodology in different varieties of rice seeds (Table 1, product #6), the authors aim to develop healthy products that are phytomelatonin-rich from rice, obtaining richness in extracts of up to 216 ng·g DW^−1^ [122,123,124].

c. Phytomelatonin from mustard seeds

Two mustard varieties (*Brassica campestris*) were studied to check the possibilities of this plant material as a supplier of extracts rich in phytomelatonin. Yellow mustard seeds contained up to 660 ng·g DW^−1^, around three-fold more than in black mustard seeds (Table 1, product #7). According to the authors, oily extracts are safe to be used as antioxidants in food supplements with interesting hypocholesterolemic and hypoglycemic activities [125,126].

d. Phytomelatonin from medicinal aromatic plants (MAPs)

d.1. Phytomelatonin from St. John’s Wort (*Hypericum perforatum*)

Interestingly, a new plant source of phytomelatonin has recently been commercialized. In this case, although we do not know the supplier or the details, the phytomelatonin-rich extracts of St. John’s wort (*Hypericum perforatum*) that show a richness of around 1% are promising if the minimum levels of characteristic components of this plant such as naphthodianthrones (hypericin and others) can be guaranteed (Table 1, product #8).

d.2. Phytomelatonin from Valerian roots

The effectiveness of valerian (*Valeriana officinalis*) roots as a mild tranquilizer in cases of generalized nervousness, insomnia, restlessness, and moderate anxiety states are well known [133]. Recently, we measured the phytomelatonin content in natural roots and commercial valerian samples to study the possible contribution of phytomelatonin as a sedative and sleep improvement agent [127]. Phytomelatonin-rich extracts from valerian roots can be obtained, but the high variability in the phytomelatonin content of raw sources (Table 2) and their limited richness will be a major disadvantage for commercial purposes (Table 1, product #9).

d.3. Other PAMs

In order to obtain plant extracts rich in phytomelatonin, we tested hundreds of plants of different origins and varieties. One of the patterns that are repeated is the great variability in the phytomelatonin contents in plant samples, influencing their origin, variety, mode of cultivation, and preservation. Table 2 shows some of the phytomelatonin contents in multiple samples of the same species and of different origin or conditioning. There is great variability in the phytomelatonin contents, which makes it extremely difficult to obtain a raw material with guarantees for commercial purposes. Likewise, our quantification data for phytomelatonin content in different plants, although they are usually within the ranges given by the literature, tend to differ considerably, being much lower than those measured by other authors (Table 2).

Since the estimated natural phytomelatonin contents in plants were not very high, we opted to produce medicinal aromatic plants (MAPs) previously elicited to increase their phytomelatonin content [129]. In this way, we have been able to obtain extracts rich in phytomelatonin with increases in content of around 60/80 times with respect to their basal endogenous content, which makes them interesting from a commercial point of view (Table 1, product #10). Thus, our plants and extracts (Bioriex product) have been characterized as containing several natural antioxidants such as phenolics, flavonoids, and carotenoids and as showing melatonin activity in vivo in a melatonin-specific bioassay that determined the ability of phytomelatonin-rich extracts to aggregate melanophores in fish were positively verified [129].

Finally, some comments should be made about the costs and prices of products containing chemical/synthetic melatonin or phytomelatonin. Melatonin from chemical synthesis is currently very cheap, with the possibility of obtaining an acceptable quality at 95% purity for around EUR 0.25–0.3 per gram (or even cheaper). On the other hand, phytomelatonin requires the use of considerable amounts of raw material, plants, or algae, thus, having to process several kg of raw material to obtain a few mg. For example, if raw plants contain 5 μg of phytomelatonin·g DW^−1^ (a considerable amount, see Table 2) from 1 kg of plants, we can obtain five pills of 1 mg of phytomelatonin. Assuming a price of about EUR 3 per kg of dried plant, the cost of 1 mg of phytomelatonin would be EUR 0.6, about 2000 times more expensive than chemical melatonin, all without taking into account the costs of handling plants and the extraction and concentration processes necessary to obtain appropriate richness in the extracts. To give a similar example with cultured algae would be impossible to understand, considering the minimal phytomelatonin content in microalgae (see above). Surprisingly, we can find cheaper phytomelatonin pills on the market than chemical melatonin pills. In this regard, some brands are honest with their customers when they are asked about phytomelatonin, answering: “… *while waiting for science to find a reliable, validated but also affordable source (of phytomelatonin), we offer you this synthetic alternative which is just as effective, but above all much cheaper!*”. Obviously, knowing whether the melatonin contained in dietary supplements is synthetic or natural would clear up many reasonable doubts about the products currently on the market. For this, accurate methods of detection and the identification of by-products that originated from the chemical synthesis of melatonin should be applied, thus, clarifying the possible adulteration and not proposing eccentric isotopic approximations to discern between natural and chemical melatonin, as we have received. The lack of control by the competent authority does not facilitate transparency in the dietary supplement sector.

## 5. Conclusions and Future Directions

Synthetic melatonin is targeted at an important market with very relevant amounts of production, consumption, and sales. Motivated mainly by the desire of the population to consume more natural dietary supplements, the possibility of obtaining melatonin from natural sources arose a few years ago. The first investigations were aimed at obtaining phytomelatonin from plants, although obtaining it from microalgae was the first product to be marketed. One of the permanent doubts about phytomelatonin, in addition to its origin, points to its possible adulteration or enrichment with melatonin and/or precursors of chemical synthesis, thus, not obtaining 100% natural products. Currently, the studies and development of technologies for obtaining melatonin from microorganism biofactories are receiving a great boost from pharmaceutical multinational companies. Of the two published approaches, the transgenic model of *E. coli* achieved better results in terms of natural melatonin production capacity than the transgenic *S. cerevisiae* model. Consumers’ acceptance of future dietary supplements containing melatonin obtained by GMOs presents a challenge for marketers. On the other hand, phytomelatonin-rich extracts from plants, even from organically grown plants, have and will have better acceptance than melatonin from GMOs. However, obtaining 100% natural phytomelatonin-rich extracts to meet market demand is currently a challenge to be won. The problems to be solved are the low levels and high variability in the natural contents of phytomelatonin in the studied plants, the expensive concentration protocols to be applied, and the possible presence of undesirable metabolites such as alkaloids, saponins, and many others. Finally, one of the most determining aspects might be that the sale of phytomelatonin (100% natural) at the same or similar price as chemical melatonin does not currently convince the most discerning consumer.

## Figures and Tables

**Figure 1 metabolites-13-00072-f001:**
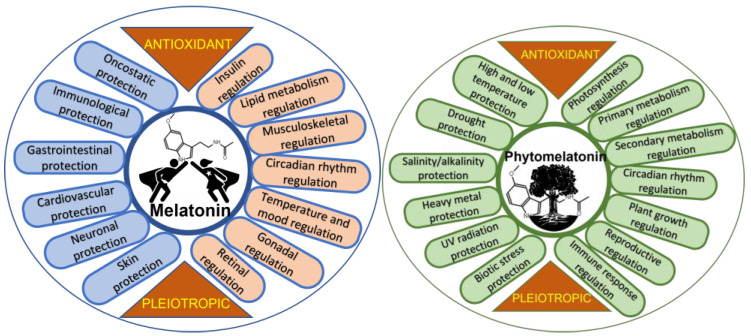
Diagram showing general roles of melatonin in humans and phytomelatonin in plants.

**Figure 2 metabolites-13-00072-f002:**
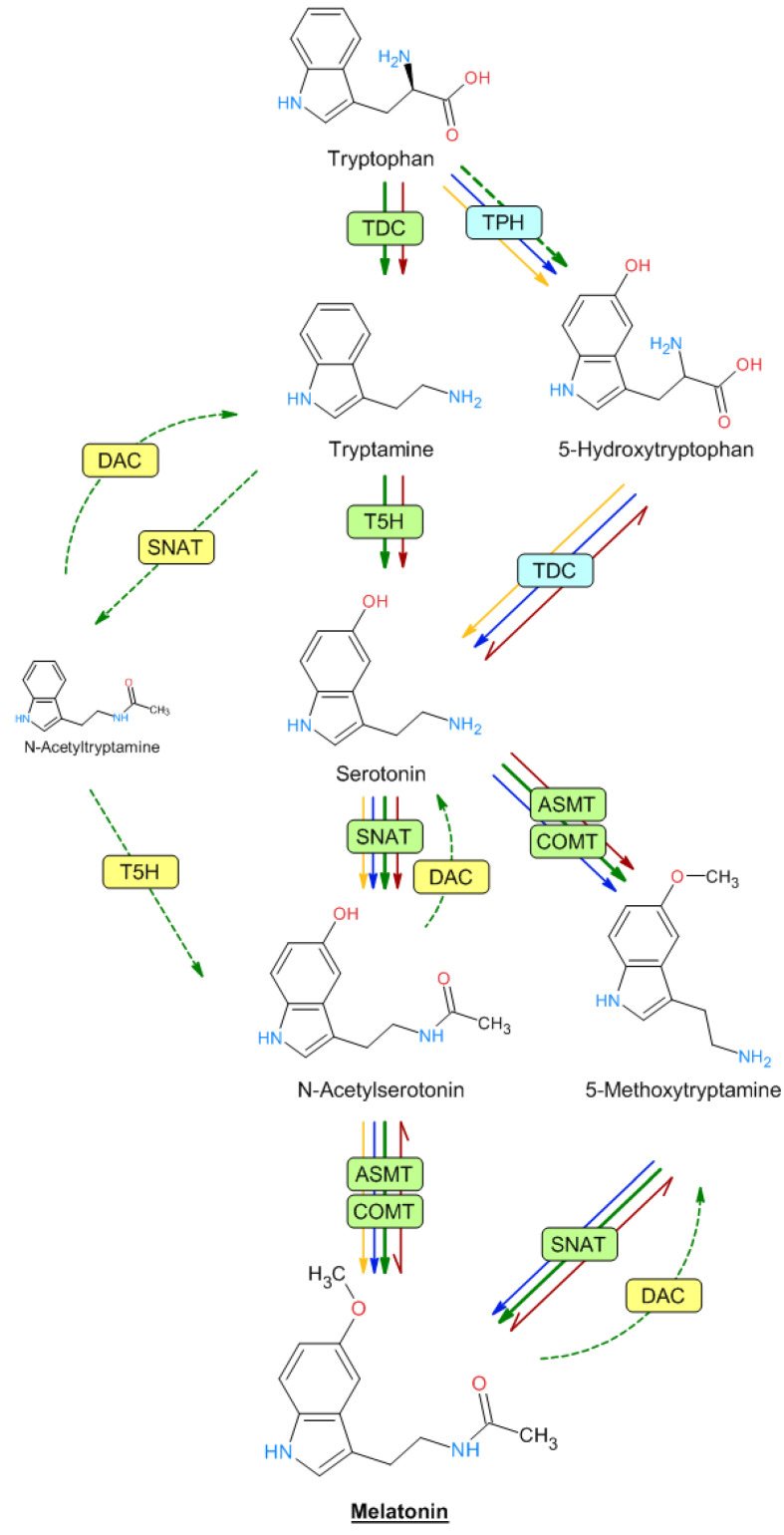
Biosynthetic melatonin pathways in mammals, plants, and microorganisms. The names of the different enzymes are described in the text. The different arrow colors denote plants (green), animals (blue), bacteria (yellow), and yeasts (red). Dashed lines indicate unproven reactions.

**Figure 3 metabolites-13-00072-f003:**
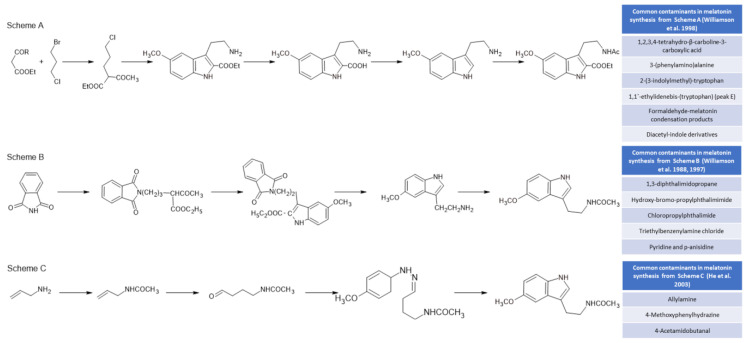
Some chemical melatonin synthesis pathways and their by-products present in synthetic melatonin preparations [101,102,103,104,105].

**Table 2 metabolites-13-00072-t002:** Phytomelatonin content of different medicinal-aromatic plants (MAPs) from different origin, sources, and harvest. Data reported in [115].

Common Name	*Scientific Name*	Organ or Zone	Phytomelatonin Content (ng/g DW ± SE)
Thyme-1	*Thymus vulgaris*	Leaf (L)	77.8 ± 3.8
Thyme-2			134.1 ± 9.3
Thyme-3			1419.5 ± 71.5
Thyme-4			625.3 ± 60.6
			26–3000 *
Lemon thyme-1	*Thymus citriodorus*	L	154.0 ± 10.1
Lemon thyme-2			243.8 ± 13.9
Red thyme-1	*Thymus zigis*	L	158.9 ± 9.9
Red thyme-2			306.9 ± 25.4
Valerian-1	*Valeriana officinalis*	Root (R)	1510.3 ± 83.2
Valerian-2			2060.8 ± 125.5
Valerian-3			60.7 ± 4.5
Valerian-4			180.5 ± 15.2
Valerian-5			455.0 ± 27.6
Valerian-6			745.9 ± 55.3
			80–300 *
Gentian-1	*Gentiana lutea*	R	222.9 ± 11.5
Gentian-2			113.7 ± 6.6
			180–300 *
Sage-1	*Salvia officinalis*	Entire(E)	17.2 ± 0.8
Sage-2		L	146.5 ± 7.7
Sage-3			223.6 ± 18.5
			32–29,000 *
Spanish sage	*Salvia lavanducifolia*	L	136.9 ± 10.6
Chamomille-1	*Matricaria chamomilla*	L	61.3 ± 2.6
Chamomille-2			95.4 ± 5.1
Lemon balm mint-1	*Melissa officinalis*	L	18.5 ± 1.3
Lemon balm mint-2			11.4 ± 1.0
Lemon balm mint-3			26.8 ± 2.1
			50–100 *
Cat’s claw-1	*Uncaria tomentosa*	E	64.5 ± 5.5
Cat’s claw-2			72.9 ± 6.9
	*Uncaria rhynchophylla*		2460 *
Lemon verbena-1	*Aloysia citriodora*	L	239.4 ± 18.3
Lemon verbena-2			250.9 ± 20.2
			1200 *
St. John’s Wort-1	*Hypericum perforatum*	R	3650.7 ± 201.6
St. John’s Wort-2			1500.3 ± 122.3
St. John’s Wort-3			2265.8 ± 138.9
			11–23,000 *
Harpagophyte	*Harpagophytum procumbens*	R	16.8 ± 1.2
MAP-1	*Different species*	L	25–125
			1800–10,000 (elicited)
MAP-2	*Different species*	L	50–230
			3200–20,000 (elicited)

* Phytomelatonin content intervals reported in the literature.

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
