# Peer review of "Melatonin from Microorganisms, Algae, and Plants as Possible Alternatives to Synthetic Melatonin"

_metabolites, 2023, doi:10.3390/metabo13010072_

Round 1

Reviewer 1 Report

I read a perfect reviewer paper. Congratulations to the authors.

Author Response

Rev #1

I read a perfect reviewer paper. Congratulations to the authors.

Response: Thank you for your flattering comment.

Reviewer 2 Report

This is an interesting paper based on the growing need of pure melatonin to treat a variety of disorders. However,  I found the description of the potential benefits of melatonin  quite overemphasized and not fully supported by the relevant references. I would suggest to revise critically this point .

Author Response

Rev #2

This is an interesting paper based on the growing need of pure melatonin to treat a variety of disorders. However, I found the description of the potential benefits of melatonin quite overemphasized and not fully supported by the relevant references. I would suggest to revise critically this point.

R: Thank you for your comment. We have incorporated a sentence in the Introduction section in this regard (lines 73-76): However, even though melatonin is a molecule that has been widely studied since the 1950s, the studies carried out require more clinical and extensive double-blind trials, in order to clarify its sometimes confusing pleiotropic action [33,34].

Reviewer 3 Report

The manuscript represents a detailed review of natural melatonin from microbial and plant origin. The review is very well organized and leads to sound and concrete conclusions.

Using the term "phytomelatonin" is somehow misleading and makes the reader think it is another chemical compound. I suggest using "natural melatonin" or just melatonin instead of "phytomelatonin".

On page 9: the paragraphs "d. Phytomelatonin from St. John's Wort" and "e. Phytomelatonin from Valerian roots" should be given under "f. Phytomelatonin from medicinal-aromatic plants (MAPs)". The heading "f. Phytomelatonin from medicinal-aromatic plants (MAPs)" should be renamed to "d. Phytomelatonin from medicinal and aromatic plants (MAPs)"

Author Response

Rev #3

The manuscript represents a detailed review of natural melatonin from microbial and plant origin. The review is very well organized and leads to sound and concrete conclusions.

R: Thanks for your comment.

Using the term "phytomelatonin" is somehow misleading and makes the reader think it is another chemical compound. I suggest using "natural melatonin" or just melatonin instead of "phytomelatonin".

R: We use the term phytomelatonin referring to melatonin of plant origin (plants and algae). This term is widely spread and is continually used in studies of phytochemistry, plant physiology, botany, etc., on melatonin in plants. We are in favor of not changing the terminology in the text, but we have reinforced this terminology with a sentence in the Introduction section (lines 100-104).

On page 9: the paragraphs "d. Phytomelatonin from St. John's Wort" and "e. Phytomelatonin from Valerian roots" should be given under "f. Phytomelatonin from medicinal-aromatic plants (MAPs)". The heading "f. Phytomelatonin from medicinal-aromatic plants (MAPs)" should be renamed to "d. Phytomelatonin from medicinal and aromatic plants (MAPs)"

R: Thank you for your comment. We have incorporated the suggested changes into the new version.

Reviewer 4 Report

The review article entitled “Melatonin from microorganisms, algae, and plants as possible alternatives to synthetic melatonin” touches a new trend topic of the chemical and biological synthesis of melatonin for human consumption, mainly as dietary supplements. The pros and cons of obtaining melatonin from microorganisms, and phyto-melatonin from plants and algae are analyzed, as well as the advantages of natural melatonin avoiding unwanted chemical byproducts from the chemical synthesis of melatonin. There’s a lot of literature available on melatonin role on different aspect but the current review is interesting and demonstrate the role of melatonin in different aspects of obtaining 100% natural phyto-melatonin rich extracts to meet market demand is currently a challenge to be won. The problems to be solved would be the low levels and high variability of the natural contents of phyto-melatonin in the studied plants, the expensive concentration protocols to be applied, and the possible presence of undesirable metabolites such as alkaloids, saponins, and many others. Finally, one of the most determining aspects might be that the sale of phyto-melatonin (100% natural) at the same or similar price as that chemical melatonin does not currently convince the most discerning consumer.

The review article is sufficiently clear. English is good and comprehensible. The subject of this work is interesting. In my opinion, the following issues should be addressed in the manuscript and considered for acceptance in the journal “Metabolites”. 

Minor suggestions:

Line 33-35; please clarify and also add the reference

Line 41. Please add the reference

Line 88; I suggest to revise the Figure 1, if possible draw an attractive figure regarding the differential role of melatonin in plant and human.

The author are need to add some latest references published in last three years; https://doi.org/10.1186/s12870-021-03160-w,  https://doi.org/10.3390/antiox11020359

Line 177-178; initially, melatonin was obtained for experimental and clinical studies from animal sources (pineal gland and urine), with the consequent risk of viral transmission. Please clarify?

Line 218; the author need to write the full name of E.Coli in the title

Line 265; rephrase the sentence

Line 293; please clearly write the name of St. John's Wort (Hypericum perforatum)

Line 309; Rephrase the sentence

Line 314; Delete “As can be seen,

Line 321-223; Rephrase the sentence

Line 345-349; Suggestion, the author need to clarify the mentioned lines and clearly describe it.

I suggest that the author can reduce the self citation.

At the end author need to thoroughly check the English of manuscript and correct all the minor spelling and sentence structure. After the required minor suggestion I recommend that this article is suitable for publication in the journal.

Congratulation to the authors

Author Response

Rev #4

The review article entitled “Melatonin from microorganisms, algae, and plants as possible alternatives to synthetic melatonin” touches a new trend topic of the chemical and biological synthesis of melatonin for human consumption, mainly as dietary supplements. The pros and cons of obtaining melatonin from microorganisms, and phyto-melatonin from plants and algae are analyzed, as well as the advantages of natural melatonin avoiding unwanted chemical byproducts from the chemical synthesis of melatonin. There’s a lot of literature available on melatonin role on different aspect but the current review is interesting and demonstrate the role of melatonin in different aspects of obtaining 100% natural phyto-melatonin rich extracts to meet market demand is currently a challenge to be won. The problems to be solved would be the low levels and high variability of the natural contents of phyto-melatonin in the studied plants, the expensive concentration protocols to be applied, and the possible presence of undesirable metabolites such as alkaloids, saponins, and many others. Finally, one of the most determining aspects might be that the sale of phyto-melatonin (100% natural) at the same or similar price as that chemical melatonin does not currently convince the most discerning consumer.

The review article is sufficiently clear. English is good and comprehensible. The subject of this work is interesting. In my opinion, the following issues should be addressed in the manuscript and considered for acceptance in the journal “Metabolites”. 

R: Thank you for your comments.

Minor suggestions:

Line 33-35; please clarify and also add the reference

R: We have incorporated a reference.

Line 41. Please add the reference

R: We have incorporated a reference.

Line 88; I suggest to revise the Figure 1, if possible draw an attractive figure regarding the differential role of melatonin in plant and human.

R: Thanks for your suggestion. However, we consider it very difficult to make a comparative figure of differential and/or similar effects of melatonin between plants and animals due to their physiological differences, among many others. We think that the effects or roles of melatonin shared by animals (humans) and plants are captured by the common labels of "Antioxidant" and "Pleiotropic".

The authors are need to add some latest references published in last three years; https://doi.org/10.1186/s12870-021-03160-w, https://doi.org/10.3390/antiox11020359

R: These refs. have been incorporated.

Line 177-178; initially, melatonin was obtained for experimental and clinical studies from animal sources (pineal gland and urine), with the consequent risk of viral transmission. Please clarify?

R: The sentence has been extended.

Line 218; the author need to write the full name of E.Coli in the title

R: Corrected.

Line 265; rephrase the sentence

R: Rewritten sentence.

Line 293; please clearly write the name of St. John's Wort (Hypericum perforatum)

R: Corrected

Line 309; Rephrase the sentence

R: Rewritten sentence.

Line 314; Delete “As can be seen,

  1. Corrected.

Line 321-323; Rephrase the sentence

R: Rewritten sentence.

Line 345-349; Suggestion, the author need to clarify the mentioned lines and clearly describe it.

R: Rewritten sentence.

I suggest that the author can reduce the self citation.

R: We have reduced our citations in 6 refs.

At the end author need to thoroughly check the English of manuscript and correct all the minor spelling and sentence structure. After the required minor suggestion I recommend that this article is suitable for publication in the journal.

Congratulation to the authors